# Commensal *Neisseria* species share immune suppressive mechanisms with *Neisseria gonorrhoeae*

**Weiyan Zhu**[1], **Maria X. Cardenas-Alvarez**[2], **Joshua Tomberg**[2], **Marguerite B. Little**[2], **Joseph A. Duncan**[1,2]*, **Robert A. Nicholas**[2,3]*

**1** Department of Medicine, Division of Infectious Diseases, University of North Carolina at Chapel Hill, Chapel Hill, NC, United States of America, **2** Department of Pharmacology, University of North Carolina at Chapel Hill, Chapel Hill, NC, United States of America, **3** Department of Microbiology & Immunology, University of North Carolina at Chapel Hill, Chapel Hill, NC, United States of America

* joseph_duncan@med.unc.edu (JAD); nicholas@med.unc.edu (RAN)

**Data Availability Statement:** All relevant data are within the manuscript and its Supporting Information files.

## Abstract

*Neisseria gonorrhoeae* is a highly adapted human sexually transmitted pathogen that can cause symptomatic infections associated with localized inflammation as well as asymptomatic and subclinical infections, particularly in females. Gonococcal infection in humans does not generate an effective immune response in most cases, which contributes to both transmission of the pathogen and reinfection after treatment. *Neisseria gonorrhoeae* is known to evade and suppress human immune responses through a variety of mechanisms. Commensal *Neisseria* species that are closely related to *N. gonorrhoeae*, such as *N. cinerea*, *N. lactamica*, *N. elongata*, *and N. mucosa*, rarely cause disease and instead asymptomatically colonize mucosal sites for prolonged periods of time without evoking clearing immunologic responses. We have shown previously that *N. gonorrhoeae* inhibits the capacity of antigen-pulsed dendritic cells to induce CD4+ T cell proliferation *in vitro*. Much of the suppressive effects of *N. gonorrhoeae* on dendritic cells can be recapitulated either by outer-membrane vesicles released from the bacteria or by purified PorB, the most abundant outer-membrane protein in *Neisseria gonorrhoeae*. We show here that three commensal *Neisseria* species, *N. cinerea*, *N. lactamica* and *N. mucosa*, show a comparable capacity to suppress dendritic cell-induced T cell proliferation *in vitro* through mechanisms similar to those demonstrated previously for *N. gonorrhoeae*, including inhibition by purified PorB. Our findings suggest that some immune-evasive properties of pathogenic *N. gonorrhoeae* are shared with commensal *Neisseria* species and may contribute to the ability of both pathogens and commensals to cause prolonged mucosal colonization in humans.

## Introduction

*N. gonorrhoeae* is the third most prevalent sexually transmitted bacterial infection worldwide, with an estimated 87 million infections per year [1]. *N. gonorrhoeae* infection is often diagnosed due to its capacity to cause robust localized inflammation that leads to copious exudate in the urethra of infected men and at the cervix of infected women. In addition to infecting the

**Funding:** JAD National Institute of Allergy and Infectious Disease U19 AI144180 R01 AI088255 RAN National Institute of Allergy and Infectious Disease U19 AI144180 ML Supported by T32 grants from the National Institute of Allergy and Infectious Disease T32AI007273 T32AI007001 The funders had no role in study design, data collection and analysis, decision to publish, or preparation of the manuscript.

**Competing interests:** The authors have declared that no competing interests exist.

genital tract, *N. gonorrhoeae* is also capable of colonizing rectal and oropharyngeal mucosa [2, 3]. Interestingly, a large percentage of infected individuals are actually asymptomatic, with epidemiological studies showing asymptomatic genital tract infections are most common in women [4]. The capacity of *N. gonorrhoeae* to cause prolonged mucosal site infection is promoted by multiple immunoevasive mechanisms [5, 6].

Commensal *Neisseria* are markedly less virulent than *N. gonorrhoeae*, generally causing long-term asymptomatic colonization in most individuals, and infrequently causing invasive opportunistic infections in immunocompromised hosts [7, 8]. As colonizers of human mucosal surfaces, commensal *Neisseria* must evade or interfere with the human immune system to promote colonization rather than immune-mediated clearance. For example, it has been reported that *N. lactamica* does not prime the development of mucosal T or B cell memory during the peak period of colonization [9]. During evolution, commensal bacteria developed the capacity to modulate host immune homeostasis through complex mechanisms, which remain poorly understood [9].

*N. gonorrhoeae* and other commensal *Neisseria* species evolved from the same ancestral bacterium and harbor highly homologous genes that likely share common physiologic functions [10, 11]. We sought to determine whether commensal *Neisseria* species use similar mechanisms to evade the human immunological response. Dendritic cells (DCs), known as "professional antigen-presenting cells", play a unique role in driving primary T cell responses. At mucosal sites, dendritic cells are critical for initiating, maintaining, and shaping T cell-mediated immune responses [12, 13]. We have shown previously that DCs treated with *N. gonorrhoeae* are unable to mediate antigen-promoted proliferation of CD4 T cells, which depends in part on the capacity of *N. gonorrhoeae* to induce DCs to release the immunosuppressive cytokine IL-10 and to upregulate inhibitory cell surface signaling molecules such as PDL1 [14]. *N. gonorrhoeae* induces these changes through the prolific release of outer-membrane vesicles (OMVs), which contain high levels of the outer membrane beta-barrel protein, PorB, that functions as a channel to facilitate diffusion of nutrients across the outer membrane [14]. Importantly, we have shown that purified PorB, when added to cells in its natively folded active state, recapitulates the inhibitory activity of OMVs, suggesting that PorB within these OMVs is a major driver of DC inhibition. In this study, we compared dendritic cell responses and functions of both OMVs and purified PorB proteins from *N. gonorrhoeae* and three related commensal *Neisseria* species, *N. cinerea*, *N. lactamica* and *N. mucosa*.

## Materials and methods

### Ethics statement

All research utilizing laboratory animals was conducted in accordance with National Institutes of Health guidelines for the care and use of laboratory animals and the University of North Carolina at Chapel Hill Institutional Animal Care and Use Committee (IACUC). The protocols were reviewed and approved by the UNC IACUC (protocol # 18–150.0 and protocol # 21–124.0). Mice used to provide primary immune cells for tissue culture experiments were euthanized before tissue harvest and did not experience pain or stress. The handling and manipulation of all animals was completed under the guidance and supervision of fully trained personnel. Mice were examined daily for signs of illness or distress, and appropriate veterinary consultation was obtained whenever necessary. If signs of illness or distress were observed, the mouse was humanely euthanized with CO2. Mice were humanely euthanized by the laboratory technician using a compressed $CO_2$ gas cylinder as per the University of North Carolina IACUC guidelines (https://policies.unc.edu/TDClient/2833/Portal/KB/ArticleDet?ID=132206)

for euthanasia of mice and rats, which follow those established by the American Veterinary Medical Association Panel on Euthanasia.

## Bacterial propagation

*N. gonorrhoeae* strain FA1090 was prepared from a predominantly Opa+ frozen stock as described previously [15]. *N. lactamica* (ATCC 23972) and *N. cinerea* (ATCC 14685) were obtained from ATCC (American Type Culture Collection, Manassas, VA), and *N. mucosa* was from the stock collections of Dr. Fred Sparling, UNC-Chapel Hill. The four *Neisseria* species were passaged on Gonococcal Base (GCB) agar and grown overnight (16–18 hours) under 5% $CO_2$ at 37˚C. Colonies were collected using a sterile cotton swab, suspended in RPMI 1640 medium containing 10% FBS, and diluted to an $OD_{600}$ = 0.18 (~$2\times10^8$ CFU/mL). Bacteria were then added to DCs at a multiplicity of infection (MOI) indicated for each experiment. The bacterial density of each inoculum was confirmed by plating serial dilutions.

## Preparation of conditioned medium

To generate conditioned medium (CM) for each species, *N. gonorrhoeae*, *N. lactamica*, *N. cinerea*, and *N. mucosa* were grown overnight on GCB plates from frozen stocks. The next morning, cells were swabbed from the plates, resuspended in pre-warmed Graver-Wade medium (GWM; [16]) at an $OD_{600}$ of 0.2, and grown in a shaking incubator at 37˚C with 5% $CO_2$. After 4 to 5 hr of growth, when the densities of the cultures reached an $OD_{600}$ of 1.0, media were collected and used to prepare CM. First, bacteria were removed by centrifugation at $2500 \times g$ for 10 minutes, then the resulting clear medium was filtered through a sterile 0.22 µm filter. A small aliquot of each sample was used for SDS-PAGE and protein staining, and the remaining amount was used to treat DC cultures. The concentrations of particles (i.e., OMVs) in CM were measured using a NanoSight NS500 (Malvern Panalytical, Ltd, UK).

## Refolding and purification of PorB proteins

The DNA sequences of the *porB* genes from the three commensal species were obtained from GenBank and used to design PCR primers complementary to the 5' and 3' ends of the coding sequences for the mature proteins (**Table 1**). The primers were designed with BamHI and EcoRI sites on the 5' and 3' oligonucleotides respectively. The resulting PCR products were digested with BamHI and EcoRI and cloned into the vector pT7-HTb digested with the same enzymes. The resulting constructs fused the 1st amino acid immediately following the signal sequence cleavage site with the HT linker, which adds a hexahistidine tag and tobacco etch protease (TEV) site to the N-terminus. After sequencing the inserts, the vectors were transformed into BL21*, and the resulting bacteria were grown at 37˚C to an $OD_{600}$ of 1 and induced overnight with 0.5 mM IPTG. The cells were pelleted, lysed with an EmulsiFlex C-5 cell disrupter (Avestin, Ottowa, CA), and spun at low speed (8000 x $g$) to pellet inclusion

**Table 1. Primers used to amplify *porB* genes from *N. cinerea*, *N. lactamica*, and *N. mucosa* genomic DNA.**

| Primer Name | Primer Sequence |
|---|---|
| 5'-Bam_NLa_por | AGA**GGATCC**GATGTTACCCTGTACGGCAC |
| 3'-Eco_NLa_por | AGA**GAATTC**TTAGAATTTGTGGCGCAGACC |
| 5'-Bam_Nci_por | AGA**GGATCC**GATGTTACCCTGTACGGTAC |
| 3'-Eco_Nci_por | AGA**GAATTC**TTAGAATTTGTGACGCAAGCC |
| 5'-Bam_Nmu_por | AGA**GGATCC**TCCGATAACGAAGCCAAAATC |
| 3'-Eco_Nmuc_por | AGA**GAATTC**TTATTGCAGGCTGTGTTCCAG |

bodies. The inclusion bodies were washed and refolded using a modification of the method described by Olesky *et al.* [17]. Briefly, inclusion bodies were solubilized in 6 M guanidine HCl, then slowly diluted into 100 mM Tris, 500 mM NaCl, 2 mM DTT, 0.5 mM EDTA, 1% LDAO, pH 8.2. After stirring overnight, the solution was clarified by centrifugation and purified on a HisTrap column (Cytiva, Marlborough, MA, USA). The purified HT-PorB proteins were pooled, concentrated, and then submitted to gel filtration on a Sephacryl S-300 column in 20 mM Tris•HCl, 150 mM NaCl, 0.1% LDAO, pH 8.0. All four porins eluted in a single peak at the size expected of a porin trimer (**Fig 1A–1D**). Fractions from the trimer peak were pooled, concentrated, and dialyzed. The pooled protein showed a single, highly pure band on SDS-polyacrylamide gel electrophoresis (SDS-PAGE) and Coomassie Brilliant Blue R-250 staining. Pools were then frozen at -80˚C until use.

## SDS-PAGE and protein staining

Samples containing 20 l of concentrated CM or 1 g of recombinant PorB from the indicated bacteria were mixed with SDS sample buffer and heated at 95C for 10 minutes. Proteins were separated in 4–12% Novex Tris-Glycine gels (Invitrogen, Eugene, Oregon), and stained with Coomassie Brilliant Blue R-250 or SYPRO ruby (Invitrogen, Eugene, Oregon) and visualized using a FluorChemE imager (Protein Simple, San Jose, CA) according to the manufacturer's protocols.

## Identification of major protein bands in OMVs by mass spectrometry

OMVs present in CM from 40 mL of culture in GWM were isolated from the four *Neisseria* species as described above and concentrated to 0.5–1 mL using a 30 kDa cutoff spin concentrator (MilliporeSigma, St. Louis, MO). Aliquots of the concentrated OMVs were solubilized in SDS-PAGE loading buffer, run on a pre-poured Any kD gel (BioRad, Hercules, CA), and stained with colloidal Coomassie Blue stain. Major bands between 30 and 50 kDa (2 for each species) were excised, alkylated with iodoacetamide, and in-gel digested with trypsin overnight at 37˚C as previously described [18]. Peptides were extracted, desalted with C18 ZipTips (MilliporeSigma, St. Louis, MO) and dried via vacuum centrifugation. The peptide samples were analyzed by LC/MS/MS using an Easy nLC 1200 coupled to a QExactive HF mass spectrometer (Thermo Fisher Scientific, Waltham, MA). Raw data files were processed individually using Proteome Discoverer version 2.5 (Thermo Fisher Scientific). Peak lists were searched using Sequest against a reviewed Uniprot *Neisseria* database and appended with a common contaminants database.

## Generation of bone marrow-derived dendritic cells

Bone marrow-derived dendritic cells (DCs) were prepared from bone marrow progenitors obtained from 6- to 12-week-old C57BL/6 mice (Jackson Laboratories, Bar Harbor, ME) as described previously [19]. Briefly, single cell suspensions of bone marrow were treated with RBC lysis buffer [150 mM $NH_4Cl$, 10 mM $KHCO_3$, and 0.1 mM $Na_2EDTA$ (pH 7.4)] and washed with PBS. Cells were cultured in RPMI 1640 medium with 10% FBS containing GM-CSF (25 ng/ml; Peprotech, Rocky Hill, NJ) and IL-4 (10 ng/ml; Peprotech, Rocky Hill, NJ). Cultures were pulsed every 48 hours with fresh medium containing GM-CSF and IL-4. After seven days in culture, immature DCs were harvested and used in T cell co-cultures as antigen-presentation cells (APCs). DCs were incubated with soluble ovalbumin (OVA; 100 μg/mL; Sigma-Aldrich, St. Louis, MO) in the presence of either 1) live bacteria (*N. gonorrhoeae*, *N. lactamica*, *N. cinerea* or *N. mucosa*) at the indicated multiplicity of infection (MOI), 2) conditioned medium (*Ng*-CM, *Nl*-CM, *Nc*-CM, or *Nm*-CM) containing equal numbers of OMVs,

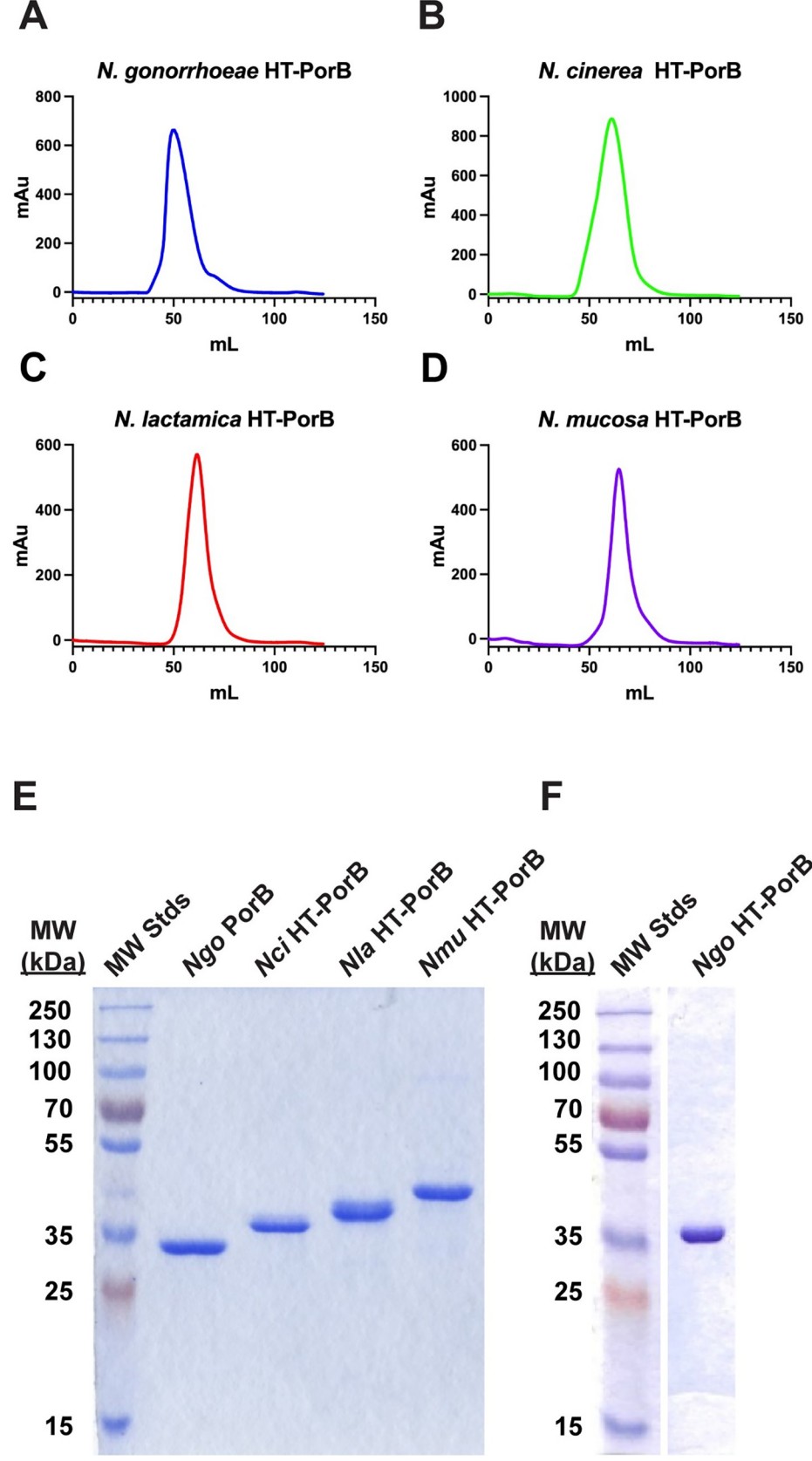

**Fig 1. Analysis of purified, refolded recombinant PorB proteins from *Neisseria* species.** (A-D) Four refolded HT-PorB preparations from the indicated *Neisseria* species were purified on a $Ni^{2+}$-NTA column, pooled and concentrated, and run on a Sephacryl S-300 column. The $OD_{260}$ was plotted versus the volume of eluent. (E) PorB proteins from four *Neisseria* species were analyzed by SDS-PAGE and stained with Coomassie Brilliant Blue R-250. The first lane is purified Ngo PorB cleaved with Tobacco Etch Virus (TEV) protease, the other lanes are purified HT-PorB from the indicated species. (F) Purity of *Ngo* HT-PorB run on a separate gel from (E). Full gels are shown in Supplemental Data.

or 3) purified HT-PorB proteins from the four *Neisseria* strains at the indicated amounts. After 24 hr, the cells were washed and then used for co-culture assays with T cells or downstream analysis.

## Evaluation of DC cytokine production and surface marker expression

DC supernatants collected 24 h after exposure were used for cytokine production. IL-6 and IL-10 levels in cell culture supernatants were determined using the OptEIA$^{TM}$ mouse IL-10 ELISA kit (BD Biosciences, San Jose, CA) according to the manufacturer's protocol. Surface antigens were examined by flow cytometry using the following anti-mouse monoclonal antibodies: FITC-labeled MHC-class II, PE-Cy5-labeled CD86, APC-Cy7-labeled CD11c, and PE-labeled CD274-PE (BD Biosciences, San Jose, CA and eBioscience, San Diego, CA). The following directly conjugated anti-mouse antibodies were used for T cell staining: CD3-labeled BV421, APC-Cy7-labeled CD4, and FITC-labeled CD8 (BD Biosciences, San Jose, CA and eBioscience, San Diego, CA). Biotin anti-mouse Vβ 5.1/5.2, biotin anti-mouse I-A[b] and APC-labeled streptavidin (BD Biosciences, San Jose, CA) were also used. Saturating amounts of antibody were used to stain approximately $1 \times 10^6$ cells in FACS Buffer (1x PBS, 1% BSA, 0.1% $NaN_3$) at a final volume of 100 μl at 25˚C for 1 hour. All samples were washed with 3 ml FACS Buffer and resuspended in 200 μl of FACS Buffer with 0.4% (w/v) paraformaldehyde. Stained samples were analyzed on a BD LSRII-SOS (BD Biosciences, Palo Alto, CA). Flow cytometry was performed in the Duke Human Vaccine Institute Flow Cytometry Facility (Durham, NC). Data from 10,000–100,000 events were acquired for each sample and saved as an FCS file that was subsequently analyzed with FlowJo software (Tree Star, Ashland, OR).

## Co-culture of DC and T cells and T cell proliferation assay

T cells were isolated from the spleens and lymph nodes of OT-II mice (C57BL/6-Tg(TcraTcrb) 425Cbn/J; Jackson Laboratories, Bar Harbor, ME) using the EasySep$^{TM}$ mouse T cell isolation kit (StemCell, Cambridge, MA) and labeled with carboxyfluoroscein succinimidyl ester (CFSE; Life Technologies) as described previously [19]. DCs were treated with live bacteria, CM, or purified HT-PorB proteins for 24 hours, then washed before adding CSFE-labeled T cells. DCs and OT-II T cells were co-cultured at a 1:10 ratio in 48-well plates (Costar, Corning, NY) at a density of $5 \times 10^4$ cells/ml DCs and $5 \times 10^5$ cells/ml CSFE-labeled T cells. On day 7, T cell proliferation was assessed by measuring CFSE fluorescence in CD4$^+$TCRV5$^+$ lymphocytes using flow cytometry on a BD LSRII-SOS (BD Biosciences). Data from 10,000–100,000 events were acquired for each sample and saved as an FCS file that was subsequently analyzed with FlowJo software (Tree Star, Ashland, OR).

## Statistical analysis

Statistical analysis was performed using Prism 8 software by GraphPad Software (La Jolla, CA, USA). The significance of differences between multiple groups was assessed using one-way

ANOVA with Dunnett's multiple comparisons test. In all cases, a $P < 0.05$ was considered statistically significant.

## Results

### Three commensal *Neisseria* species, *N. cinerea*, *N. lactamica* and *N. mucosa*, inhibit dendritic cell-induced antigen-specific T cell proliferation

To assess the effects of commensal *Neisseria* species on the DC-mediated, antigen-specific induction of T cell proliferation, bone marrow-derived DCs were pulsed with ovalbumin (OVA) and either fresh media or media containing *N. gonorrhoeae*, *N. cinerea*, *N. lactamica* or *N. mucosa* bacteria at an MOI = 10. After 24 hours, the DCs were washed and then co-cultured with naïve CFSE-labeled T cells isolated from spleens and lymph nodes of OT-II mice. After seven days, the dilution of CFSE fluorescence in the T cell population of each co-culture, which represents the degree of proliferation of the T cells, was quantified by flow cytometry. Treatment of DCs for 24 h with either *N. gonorrhoeae* or the three commensal *Neisseria* species resulted in a marked reduction of proliferation of co-cultured T cells compared to OVA-treated DCs (**Fig 2A**). The treatments all showed statistical significance compared to no bacteria, but there was no significant statistical difference between the degree of inhibition of proliferation by any of the *Neisseria* species compared to each other (**Fig 2B**). These results demonstrate that these non-pathogenic *Neisseria* species have the same capacity to suppress dendritic cell-induced, antigen-specific T cell proliferation as we have demonstrated previously for *N. gonorrhoeae*.

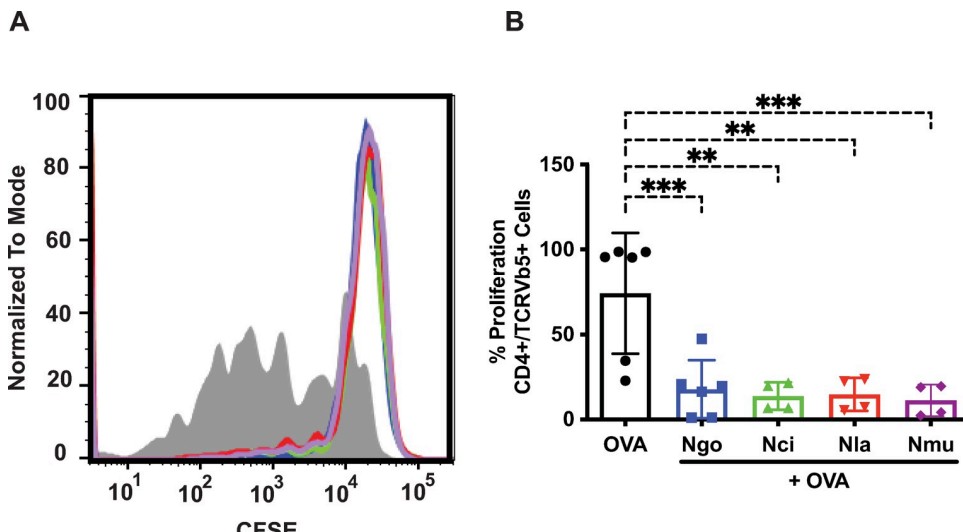

**Fig 2. *N. gonorrhoeae* and three commensal *Neisseria* species inhibit OVA-primed dendritic cell-induced T cell proliferation.** Dendritic cells (DCs) were exposed to *N. gonorrhoeae* (*Ngo*), *N. cinerea* (*Nci*), *N. lactamica* (*Nla*), or *N. mucosa (Nmu)* bacteria at an MOI = 10 in the presence of OVA for 24 hours. DCs were then washed and co-cultured with CFSE-loaded OT-II T cells for seven days. T cell proliferation stimulated by DC presentation of OVA was assessed by measuring dilution of CFSE fluorescence using flow cytometry as detailed in Materials and Methods. (A) Representative overlay histograms of CFSE fluorescence in OVA-directed CD4+ T cells proliferation after incubation with DCs plus OVA alone (filled grey), *Ngo* plus OVA (blue), *Nci* plus OVA (green), *Nla* plus OVA (red), or *Nmu* plus OVA (purple). (B) Average of the mean percentage of CD4+ T cells that proliferated under each condition from at least three separate experiments are shown. Data were analyzed using one-way ANOVA with Dunnett's multiple comparisons test (*, $P<0.05$; **, $P<0.01$; ***, $P<0.001$; and ****, $P<0.0001$).

### Conditioned medium from cultures of commensal *Neisseria* species recapitulates the inhibition of DC-induced antigen-specific T cell proliferation

Gram-negative bacteria naturally elaborate outer membrane vesicles (OMVs) into the extracellular milieu during growth, which can influence signaling pathways in host cells exposed to the bacteria. We have shown previously that conditioned medium (CM) from cultures of *N. gonorrhoeae* contains large quantities of OMVs and that a >100 kDa retentate fraction of CM containing OMVs recapitulates the inhibitory effect of *N. gonorrhoeae* bacteria on dendritic cells. To test whether OMVs in CM from commensal *Neisseria* species also inhibit DC-induced T cell proliferation, we generated CM from cultures of *N. gonorrhoeae*, *N. cinerea*, *N. lactamica* and *N. mucosa* as described in Materials and Methods. When added together with OVA to DCs for 24 h before washing and incubating with CFSE-labeled OVA-directed CD4+ T cells, CM containing similar numbers of OMV particles from the three cultures inhibited T cell proliferation to a comparable extent as live bacteria (**Fig 3A**). When submitted to SDS-PAGE, a variety of bacterial proteins were detected in CM from each species (**Fig 3B**), including major bands at ~35–40 kDa that were confirmed by mass spectrometry to represent PorB monomer (the bands in **Fig 3B** indicated with an arrow were verified to be the species-specific PorB homolog by mass spectrometry; see Materials and Methods). Optical microscopy-based nanoparticle analysis of the CM samples revealed particles with sizes ranging from 109.4 nm to 167.3 nm, consistent with the presence of OMVs (**Table 2**). Taken together, these results

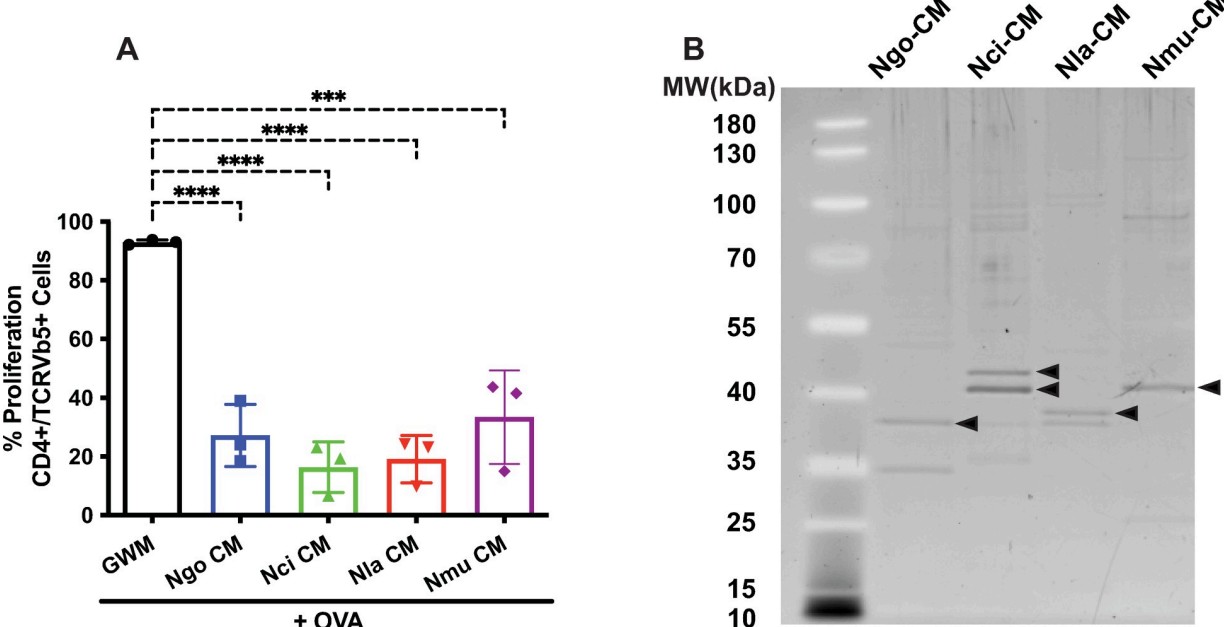

**Fig 3. Conditioned medium prepared from cultures of commensal *Neisseria* species inhibits OVA-primed DC-induced T cell proliferation.** Aliquots of conditioned medium (CM) of the different bacteria (*Ngo*-CM, *Nci*-CM, *Nla*-CM, or *Nmu*-CM) prepared from cultures grown in Grave-Wade medium (GWM) containing equivalent numbers of particles (**Table 2**) were added to DCs in the presence of OVA for 24 hours as indicated. DCs were then co-cultured with CFSE-loaded OT-II T cells for seven days. (A) Mean percentage of CD4+ T cell proliferation stimulated by dendritic cell presentation of GWM plus OVA (black), *Ngo*-CM plus OVA (blue), *Nci*-CM plus OVA (green), *Nla*-CM plus OVA (red), or *Nmu*-CM plus OVA (purple). Data were analyzed using one-way ANOVA with Dunnett's multiple comparisons test (*, $P<0.05$; **, $P<0.01$; ***, $P<0.001$; and ****, $P<0.0001$). (B) Conditioned media prepared from liquid cultures of the 4 *Neisseria* species were loaded on SDS-PAGE gel with $1.5 \times 10^6$ particles per lane and stained and visualized by SYPRO Ruby. Bands marked with an arrow (➤) were identified by Mass Spectrometry as the PorB homolog of the species from which the CM was prepared. Both bands from Nci-CM were identified as PorB.

**Table 2. Mean particle sizes and concentrations of OMVs in CM from the four *Neisseria* species.**

| Conditioned Media | | |
|---|---|---|
| Species | Mean Particle Size +/- SD (nm) | Mean Particle Concentration +/- SD (particles/mL) |
| *N. gonorrhoeae* FA1090 | 109.4 ± 0.8 | $4.37 \times 10^{10} \pm 1.55 \times 10^{9}$ |
| *N. cinerea* | 146.6 ± 4.4 | $6.38 \times 10^{10} \pm 2.92 \times 10^{9}$ |
| *N. lactamica* 972 | 167.3 ± 5.0 | $5.88 \times 10^{10} \pm 3.11 \times 10^{9}$ |
| *N. mucosa* | 159.2 ± 3.3 | $4.67 \times 10^{10} \pm 1.59 \times 10^{9}$ |

demonstrate that both live bacteria and CM from *in vitro* growth of the commensal *Neisseria* species, *N. cinerea*, *N. lactamica* and *N. mucosa*, are similarly effective at suppressing dendritic cell-mediated T cell proliferation as that shown previously for *N. gonorrhoeae*. Furthermore, these commensal species, much like *N. gonorrhoeae*, elaborate copious amounts of OMVs, which likely mediate the suppressive effects of CM from *in vitro* cultures.

## Live bacteria and CM from commensal *Neisseria* species increase the expression of the immunosuppressive molecules IL-10 and PDL1 in dendritic cells

Our prior studies showed that *N. gonorrhoeae* stimulation of dendritic cells led to an increase in expression of the immunosuppressive molecules IL-10 and the cell-surface ligand, PDL1. We tested whether *N. cinerea*, *N. lactamica*, and *N. mucosa* bacteria or CM would induce similar increases in immunosuppressive cytokine production in DCs. The quantity of IL-10 from the supernatants of dendritic cells was measured by ELISA 24 hr after exposure to the three *Neisseria* species at an MOI = 1. We found that all three *Neisseria* species induced similar, robust IL-10 secretion in DC, no matter whether the dendritic cells were exposed to live bacteria (**Fig 4A**) or to OMV-containing CM (**Fig 4B**). Likewise, DCs from these same cultures also

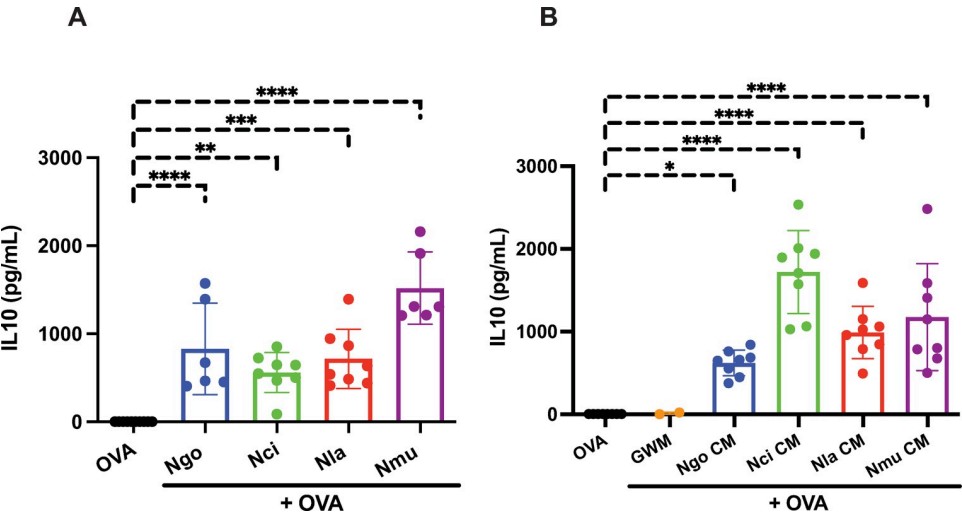

**Fig 4. Live bacteria and conditioned medium prepared from four *Neisseria* species up-regulated IL10 secretion in DCs.** Dendritic cells were exposed to **(A)** whole live bacteria (*N. gonorrhoeae* (*Ngo*), *N. lactamica* (*Nla*), *N. cinerea* (*Nci*), or *N. mucosa* (*Nmu*)), or **(B)** conditioned medium (*Ng*-CM, *Nla*-CM, *Nci*-CM, or *Nmu*-CM) in the presence of OVA for 24 hours, at which time IL-10 secretion was measured using ELISA. Bar graphs are shown as mean concentrations of IL10 (pg/ml) ± standard deviation with individual points. Statistical significance was determined by one-way ANOVA with Dunnett analysis for multiple comparisons (*, $P<0.05$; **, $P<0.01$; ***, $P<0.001$; and ****, $P<0.0001$).

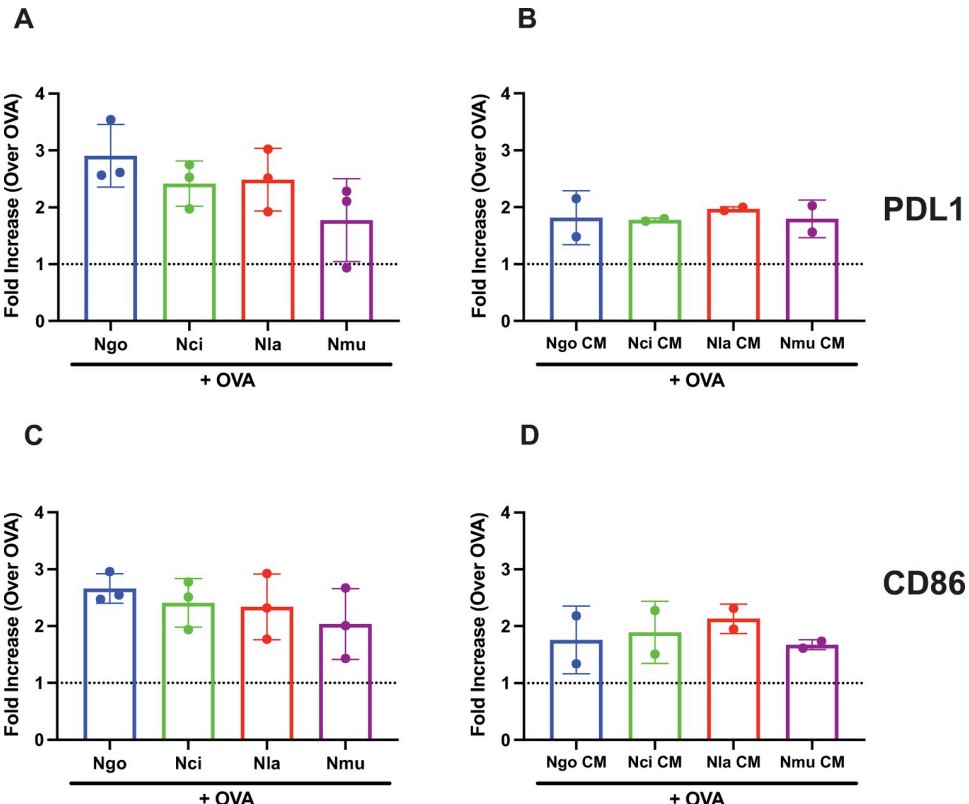

**Fig 5. Live bacteria and conditioned medium prepared from four *Neisseria* species up-regulated expression of dendritic cell surface markers.** Dendritic cells were exposed to whole live bacteria (A, B) or conditioned medium (C, D) in the presence of OVA for 24 hours and the expression of PDL1 (A, C) and CD86 (B, D) were quantified by flow cytometric analysis. The bar graphs represent the fold change in mean fluorescence intensity (MFI) for each condition compared to OVA-treatment alone. *Ngo* (bacteria or CM) + OVA (blue), *Nci* (bacteria or CM) + OVA (green), *Nmu* (bacteria or CM) + OVA (yellow), or *Nla* (bacteria or CM) + OVA (red). The fold increase in MFI was calculated as the mean of 2 independent experiments.

demonstrated increased surface expression of PDL1 and the dendritic cell activation marker CD86 in response to live bacteria (**Fig 5A and 5C**) or CM from bacterial cultures (**Fig 5B and 5D**). We conclude from these studies that the commensal *Neisseria* species exert similar immune suppressive effects on host dendritic cells as the highly related pathogen, *N. gonorrhoeae*.

## Purified *Neisseria* porins from the commensals *N. cinerea*, *N. lactamica* and *N. mucosa* suppress dendritic cell-induced T cell proliferation

In *Neisseria gonorrhoeae*, PorB is an essential outer membrane porin that acts as a nutrient channel and is one of the most abundant outer membrane proteins in the bacteria. CM from cultures of the commensal *Neisseria* species each contained a major protein band (indicated by a ◄) that was shown by mass spectrometry analysis to be identical to the PorB protein from the respective species (**Fig 3B**). We have shown previously that highly purified, functionally relevant *N. gonorrhoeae* PorB produced as inclusion bodies in *E. coli* and refolded *in vitro* suppresses DC-induced T cell proliferation [14]. The PorB homologs in the four species share between 23–80% identity with each other (**Table 3**). Compared to PorB from *N. gonorrhoeae*, PorB from *N. lactamica* is the most homologous (80%) and PorB from *N. mucosa* is the least

**Table 3. Sequence identity of different *Neisseria* PorB homologs.**

|  | *N. cinerea* | *N. lactamica* | *N. mucosa* |
| --- | --- | --- | --- |
| *N. gonorrhoeae* | 58% | 80% | 24% |
| *N. cinerea* | - - - | 64% | 27% |
| *N. lactamica* | - - - | - - - | 23% |

(24%). Indeed, PorB from *N. mucosa* was the most distant from the PorB orthologs of the other species. A multi-sequence alignment of the four PorB orthologs (**Fig 6**) shows that the regions of highest identity correspond to the membrane-embedded beta-barrel strands of the PorB structure.

To determine if the PorB orthologs from the commensal species were as inhibitory as PorB from *N. gonorrhoeae*, we prepared highly purified, refolded HT-PorB trimers from *N. gonorrhoeae* and the commensal *Neisseria*, *N. cinerea*, *N. lactamica* and *N. mucosa* (**Fig 1**; see Materials and Methods for details). We showed previously that the presence of an HT tag at the N-terminus of recombinant *Ngo* PorB had no effect on its capacity to inhibit DC-mediated T cell proliferation [14]. To assess whether the refolded, recombinant PorB porins (HT-PorB) from *N. cinerea*, *N. lactamica*, and *N. mucosa* exerted similar inhibitory effects on dendritic cells as *N. gonorrhoeae* HT-PorB, we incubated DCs with HT-PorB proteins from each *Neisseria* species (at a final detergent concentration of 0.002% LDAO) in the presence of OVA antigen for 24 h, and quantified DC-induced antigen-specific T cell proliferation after 7 days of co-culture as described above. Each HT-PorB preparation showed a dose-dependent capacity to suppress DC-mediated T cell proliferation (**Fig 7**).

## Discussion

Gram-negative *Neisseria* contain ten human-restricted species. Two species, *N. gonorrhoeae* and *N. meningitidis*, have evolved to become pathogens that cause disease in humans. Other species, including *N. cinerea*, *N. lactamica*, *and N. mucosa*, are considered commensals of the human nasopharynx. One distinguishing feature between pathogenic and non-pathogenic *Neisseria* is the capacity of pathogenic *Neisseria* to induce inflammatory responses (at least in a subset of patients), including inflammatory cytokine production and PMN influx to the site of infection [20]. One known virulence factor associated with the pathogenic *Neisseria* species is lipooligosaccharide. Pathogenic *Neisseria* express a lipid A phosphoethanolamine transferase, which generates a lipooligosaccaharide with enhanced stimulation of host TLR4 relative to unmodified LOS [21, 22]. The capacity of pathogenic and commensal *Neisseria* bacteria to induce adaptive immune responses is largely unstudied. However, adaptive immune responses, measured by the presence of antibodies directed against the bacteria, are weak against both the commensal *N. lactamica* and the pathogen *N. gonorrhoeae*, which both colonize human mucosal surfaces [9]. All commensal bacteria (including *Neisseria*) have co-evolved with humans to effectively colonize mucosal sites and as a result exert multiple mechanisms to modulate immune response and escape clearance [23]. As a key player in regulating immunity and maintenance of mucosal homeostasis, DCs, which recognize pathogen-associated molecular patterns (PAMPs) through pattern recognition receptors (PRRs) such as Toll-like receptors (TLRs) and NOD-like receptors (NLRs), shape the course of an immune response by the secretion of pro-inflammatory cytokines and chemokines [24, 25]. In this study, we showed that both pathogenic *N. gonorrhoeae*, as well as non-pathogenic *N. lactamica*, *N. cinerea*, and *N. mucosa*, induce a tolerogenic phenotype in DCs that is characterized by IL-10 secretion and increased PDL1 expression. Among the cytokines produced by DCs, IL-10 is

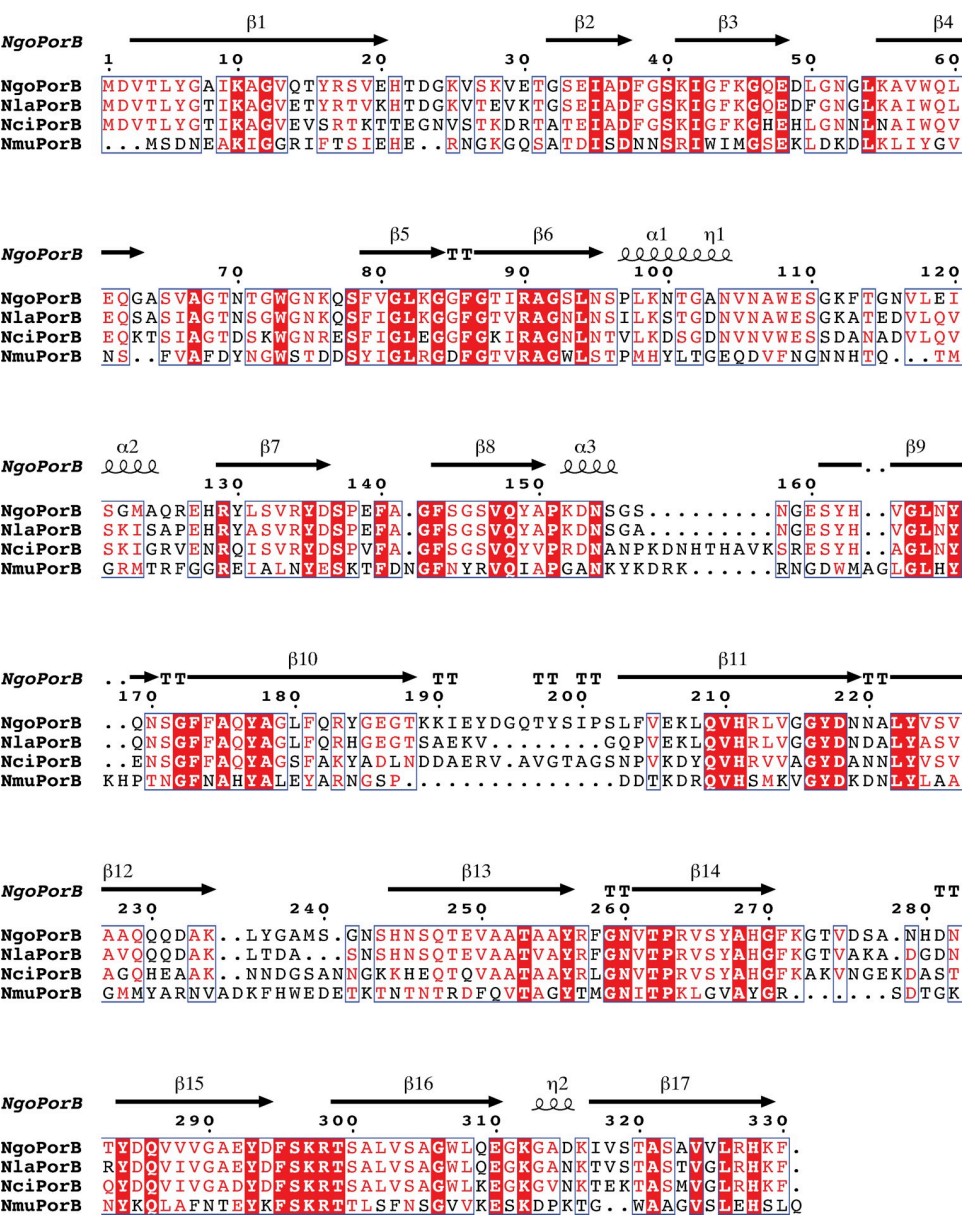

**Fig 6. Amino acid sequence alignment of the PorB homologs from *N. gonorrhoeae*, *N. lactamica*, *N. cinerea*, and *N. mucosa*.** Each of the four *Neisseria* genomes examined encode a PorB homolog. An alignment depicting the homology between PorB at the primary amino acid sequence structure from four species of *Neisseria* is shown. Amino acids identical among the four PorB homologs are shown in a dark red box. Amino acids corresponding to the beta-barrel regions of PorB using *N. meningitidis* PorB as a structure model are indicated as arrows above the sequence, alpha helical segments are denoted with a helix above the sequence, and the regions marked with T represent turns. This figure was prepared using ESPript 3.0 (https://espript.ibcp.fr/ESPript/cgi-bin/ESPript.cgi) [41].

a key regulatory cytokine limiting and ultimately terminating excessive T-cell responses to microbial pathogens, preventing chronic inflammation and tissue damage [26]. Murine models of *N. gonorrhoeae* vaginal infection suggest that host IL-10 production in response to infection plays a role in blunting adaptive immune responses to *N. gonorrhoeae* bacteria [19, 27, 28]. Our data suggest that promoting DCs to elicit an IL-10 predominant response could be a common feature of *Neisseria* species that reduces adaptive immunity to the bacteria.

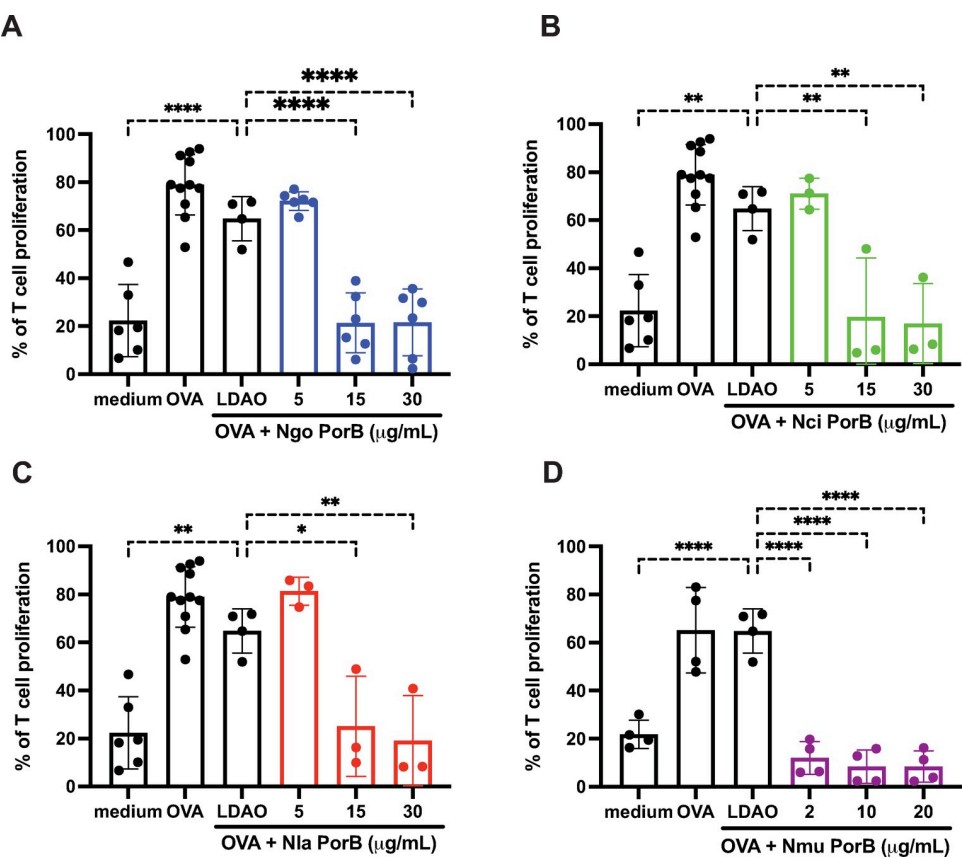

**Fig 7. Purified recombinant PorB inhibits OVA-pulsed dendritic cell-induced T cell proliferation.** DCs were exposed to OVA with LDAO-containing buffer (vehicle) or PorB proteins from the indicated species (A, *Ngo*; B, *Nla*; C, *Nci*; and D, *Nmu*) at three different concentrations. The DCs were subsequently incubated with CFSE-labeled OT-II T cells and T cell proliferation was assessed after seven days by flow cytometry as described in Materials and Methods. Bar graphs show combined results of T cell proliferation assays assessed in three independent experiments with the indicated dose of each PorB. Statistical significance was determined by one-way ANOVA with Dunnett analysis for multiple comparisons against the vehicle control (*, $P<0.05$; **, $P<0.01$; ***, $P<0.001$; and ****, $P<0.0001$). LDAO, Lauryldimethylamine oxide.

Prolific blebbing of OMVs is a characteristic of *Neisseria* species; however, the consequences of this process are not fully understood. There is a growing recognition that OMVs from Gram-negative bacterial species have immunomodulatory effects on host organisms [29]. For example, DCs can internalize OMVs from the commensal *Bacteroides fragilis* and induce IL-10-producing CD4$^+$Foxp3$^+$ T$_{reg}$ cells through TLR2 activation by capsular polysaccharide [30]. Similar immune modulating activities have been observed with other gastrointestinal tract commensal organisms, including *Odoribacter splanchnicus* and *E. coli* [31, 32]. Our studies demonstrate that an OMV-containing fraction of CM from *N. gonorrhoeae* and now from three commensal *Neisseria* species are also capable of inducing immune inhibitory IL-10 in dendritic cells and suppressing the capacity for antigen-stimulated T cell proliferation by these cells [14]. These findings are consistent with reports by others on the effects of OMVs from *N. meningitidis* [33]. Overall, our findings indicate that OMVs shed by both pathogenic and commensal *Neisseria* species likely produce immunomodulation in hosts that parallels the effects of OMVs from other bacterial species.

In addition to stimulating IL-10 production by DCs, *N. gonorrhoeae*, *N. cinerea*, *N. lactamica* and *N. mucosa* release large quantities of porins associated with OMVs. We show here

that recombinant, refolded porins from each of these *Neisseria* species inhibit DC-primed anti-gen specific CD4+ T cell proliferation *in vitro*. The signaling mechanisms underlying porin-mediated immune suppression remain to be determined. Given the diversity of primary amino acids sequences of these proteins within the individual *Neisseria* species as well as across the genus, it is unlikely there is a single receptor-mediated signaling function across these pro-teins. PorB homologs from *N. gonorrhoeae*, *N. meningitidis*, and *N. lactamica* have been found to induce inflammatory responses that can contribute to immunostimulatory adjuvant-like properties through activation of TLR2 [34–36]. Despite this known proinflammatory signaling induced by PorB, immunization of mice with OMVs prepared from cultures of *N. meningitidis* harboring deletions in both PorA and PorB was shown recently to produce equal or superior cross-reactive bactericidal antibodies compared to OMVs prepared from *N. meningitidis* cul-tures expressing both PorA and PorB [37]. This finding is consistent with PorB in OMVs inter-fering with protective immune responses. Interestingly, porins from *N. gonorrhoeae* and *N. meningitidis* have been found to localize in host mitochondrial membranes after translocation from OMVs or when exogenously expressed by transfection [38, 39]. This mitochondrial local-ization has been found to induce apoptosis in host cells, which could contribute to the immune suppressive activity of the proteins.

Prior studies have suggested *N. mucosa* porin failed to localize to mitochondria when expressed by transfection in mammalian cells [40]. Our current studies demonstrate that prop-erly folded, recombinant *N. mucosa* porin exhibits the same immunosuppressive properties on mammalian DCs as *N. gonorrhoeae* PorB. Whether proper folding is required for the mito-chondrial localization of *Neisseria* porins reported by others has not been determined, but our findings are consistent with that possibility.

Whether porins from bacterial species other than those of the genus *Neisseria* carried by OMVs to host immune cells also have immune suppressive features is an open question. Given the findings that a diverse range of *Neisseria* porins all demonstrate immune suppressive capacity, it is a reasonable possibility that both the quantity of OMVs shed and the density of bacterial porins on those OMVs could contribute to suppression of host immune cells by Gram-negative commensal bacteria.

In conclusion, our data show that commensal *Neisseria* species share immunosuppressive properties that have recently been shown in the pathogenic *N. gonorrhoeae* and in other com-mensal bacteria [14, 22, 31, 32]. This immunosuppression occurs with live bacteria, OMVs shed by the bacteria during cell growth, or with purified recombinant PorB proteins (the most abundant protein in OMVs) from the respective *Neisseria* species. Furthermore, these com-mensal *Neisseria* share the same immune suppressive mechanisms we have previously identi-fied in *N. gonorrhoeae*, including upregulation of host immune suppressive molecules and release of suppression-inducing factors into the host environment, including highly abundant porin proteins contained in OMVs. Overall, this report suggests that immune evasion by *N. gonorrhoeae* is likely a vestige of evolution from a common commensal ancestor rather than an acquired virulence mechanism.

## Supporting information

**S1 Fig. Full gel of Fig 1E.** All of the PorB protein homologs except Ngo PorB have the 28 amino acid HTb tag at their N-terminus (designated HTb-PorB). The lanes are (left to right) Page Ruler Plus MW standards; Ngo PorB, in which the N-terminal HTb linker was cleaved off with TEV protease; HTb-Nmu PorB; HTb-Nla PorB; HTb-Nci PorB; and HTb-mouse Voltage Dependent Anion Channel (mVDAC). All of these proteins were refolded, purified on a Ni2+-NTA column, and then run on an S-300 gel filtration column. The proteins were from

the pooled fractions from gel filtration. X, lane not included in Fig 1E. The gel was dried and scanned as a PDF image.
(PDF)

**S2 Fig. Full gel of Fig 1F.** Refolding and purification of HT-PorB from N. mucosa expressed as inclusion bodies (IBs) in E. coli. The lanes are (left to right) Page Ruler Plus MW standards; Inclusion bodies of HT-Nmu PorB; Refolded Nmu HT-PorB; Filtered refolded Nmu HT-PorB; 15 mM imidazole bump from Ni2+-NTA column; refolded Ngo HT-PorB purified on a Ni2 +-NTA column followed by gel filtration on an S-300 column; 1, 2, and 3 μL of the peak pooled fractions of Nmu HT-PorB following imidazole elution from the Ni2+-NTA column. Note the slight size increase in Ngo HT-PorB in this gel compared to TEV-cleaved Ngo PorB in S1 Fig. The HT tag has no effect on the capacity of Ngo PorB to inhibit DC-mediated T cell proliferation (Ref [14]). X, lane not included in Fig 1F. The gel was dried and scanned as a PDF image.
(PDF)

**S3 Fig. Full gel of Fig 3.** Conditioned medium prepared from cultures of commensal Neisseria species submitted to SDS-PAGE. Conditioned medium from cultures of the four Neisseria species was collected, and the number of particles (i.e. OMVs) was determined as described in Materials and Methods. Equal numbers of particles (left 4 lanes) or equal protein (right 4 lanes) were mixed with SDS-PAGE loading buffer and submitted to electrophoresis. The gel was stained with Sypro Ruby and imaged on a ChemiDoc Touch Imager (BioRad). The arrows indicate the PorB band in each species as determined by mass spectrometry (see Materials and Methods for details). X, lane not included in Fig 3.
(PDF)

## Acknowledgments

We would like to thank the Michael Hooker Proteomics Facility at the University of North Carolina for the mass spectrometry experiments, and the Duke University Flow Cytometry Core Facility for providing instrument time for the T cell proliferation and surface marker analyses. We also thank other members of the Duncan and Nicholas labs for advice and encouragement.

## Author Contributions

**Conceptualization:** Weiyan Zhu, Joseph A. Duncan, Robert A. Nicholas.

**Data curation:** Weiyan Zhu, Maria X. Cardenas-Alvarez, Joshua Tomberg, Joseph A. Duncan, Robert A. Nicholas.

**Formal analysis:** Weiyan Zhu, Joseph A. Duncan, Robert A. Nicholas.

**Funding acquisition:** Joseph A. Duncan, Robert A. Nicholas.

**Investigation:** Weiyan Zhu, Maria X. Cardenas-Alvarez, Joshua Tomberg, Marguerite B. Little.

**Methodology:** Weiyan Zhu, Maria X. Cardenas-Alvarez, Joshua Tomberg, Marguerite B. Little, Joseph A. Duncan, Robert A. Nicholas.

**Project administration:** Joseph A. Duncan, Robert A. Nicholas.

**Resources:** Joseph A. Duncan, Robert A. Nicholas.

**Supervision:** Joseph A. Duncan, Robert A. Nicholas.

**Validation:** Weiyan Zhu, Joshua Tomberg, Joseph A. Duncan, Robert A. Nicholas.

**Visualization:** Weiyan Zhu, Maria X. Cardenas-Alvarez, Joshua Tomberg, Joseph A. Duncan, Robert A. Nicholas.

**Writing – original draft:** Weiyan Zhu, Joseph A. Duncan, Robert A. Nicholas.

**Writing – review & editing:** Weiyan Zhu, Maria X. Cardenas-Alvarez, Marguerite B. Little, Joseph A. Duncan, Robert A. Nicholas.

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
