## [Editor Report · Decision Letter 0]

8 Mar 2023

PONE-D-23-03331Commensal Neisseria species share immune suppressive mechanisms with *Neisseria gonorrhoeae*PLOS ONE

Dear Dr. Nicholas,

Thank you for submitting your manuscript to PLOS ONE. After careful consideration, we feel that it has merit but does not fully meet PLOS ONE’s publication criteria as it currently stands. Therefore, we invite you to submit a revised version of the manuscript that addresses the points raised during the review process. Thank you for re-submitting your manuscript. The manuscript, the original reviewer's comments and the minor changes made to the manuscript in response to their reviews have been evaluated. I agree with the original decision that this is an important and interesting piece of work which is of interest to the field. During the re-review process a few minor revisions were identified that are required to clarify the methods used and to correct a statement made in the manuscript's discussion. 

1. Methods (line 152-153): were the DCs pulsed with OVA before combination with OT-II CD4+ T cells? Indicate source of OVA, concentration used/incubation time.

2. Methods (line 177-178): What ratio of DC:CD4+ responder cells was used for co-culture assays?

3. Line 323: this sentence requires modification as it implies that reference number 37 includes a Gonorrhoea challenge study.

We look forward to receiving your revised manuscript.

Kind regards,

Fiona J Radcliff

Academic Editor

PLOS ONE

Journal Requirements:

**Additional Editor Comments:**

Apologies that this feedback has taken a while to reach you. It took a little while to identify the best way forward with this manuscript. It's an unusual situation given it had already been through the system and received positive reviews. It was unable to go back to the original editor. 

---

## [Author Response · Author response to Decision Letter 0]

14 Mar 2023

Dear Dr. Radcliff:

Thank you for your email on our submission [PONE-D-23-03331], with an editorial decision of minor revision. We have edited the manuscript to address the comments raised in the reviews, and our changes are shown below for each point raised.

Point 1:

Methods (line 152-153): were the DCs pulsed with OVA before combination with OT-II CD4+ T cells? Indicate source of OVA, concentration used/incubation time.

o We agree that the methods as previously written were not clear on these points. The DCs were pulsed for 24 hr with OVA in the presence of either live bacteria, conditioned medium, or purified PorB proteins. The cells were then washed to remove these and then incubated with the OT-II CD4+ T cells for 7 days. We have clarified this point in the revised manuscript and included all the requested information. 

Point 2:

Methods (line 177-178): What ratio of DC:CD4+ responder cells was used for co-culture assays?

o The ratio of DC:CD4+ responder cells was 1:10 in the co-culture assays. We have included additional information to clarify the methods used.

Point 3:

Line 323: this sentence requires modification as it implies that reference number 37 includes a Gonorrhoea challenge study.

o We thank the reviewer for catching this mis-statement; the study indeed did not include an Ng challenge study, and instead was looking at the cross-reactive bactericidal Abs produced by immunized mice. We have modified the original sentence to reflect this.

I am also including our response to a subsequent email to address two additional points before the article will be considered for publication. The points of the email are below, along with our responses.

To comply with PLOS ONE submissions requirements, in your Methods section, please provide additional information regarding the experiments involving animals and ensure you have included details on (1) methods of sacrifice, (2) methods of anesthesia and/or analgesia, and (3) efforts to alleviate suffering.

o We have included additional information regarding the points above in the Ethics statement at the beginning of the Methods section.

2. Please note that PLOS ONE requires submissions reporting blot or gel data to comply in full with the reporting requirements described at https://journals.plos.org/plosone/s/figures#loc-blot-and-gel-reporting-requirements. We now require authors to provide the original unadjusted and uncropped images for any blot or gel data reported in PLOS ONE submissions. In our internal checks for your submission, we noted that you did not provide original raw image files supporting blot/gel data in response to our previous request.

o Please note that we already provided the requested raw images in the Supplemental Information. However, it was apparently not named as requested (S1_raw_images). Therefore, we remade the document in Adobe Illustrator and renamed it S1_raw_images.pdf.

The revised manuscript (both tracked and clean) containing all of the above changes have now been uploaded. I have also removed Supplemental Information (since that file is now redundant with the new file S1_raw_images.pdf) and uploaded S1_raw_images.pdf in its place.

---

## [Editor Report · Decision Letter 1]

22 Mar 2023

Commensal Neisseria species share immune suppressive mechanisms with *Neisseria gonorrhoeae*

PONE-D-23-03331R1

Dear Dr. Nicholas,

We’re pleased to inform you that your manuscript has been judged scientifically suitable for publication and will be formally accepted for publication once it meets all outstanding technical requirements.

Kind regards,

Fiona J Radcliff

Academic Editor

PLOS ONE
---

## [Editor Report · Acceptance letter]

30 Mar 2023

PONE-D-23-03331R1 

Commensal *Neisseria* species share immune suppressive mechanisms with *Neisseria gonorrhoeae*

Dear Dr. Nicholas:

I'm pleased to inform you that your manuscript has been deemed suitable for publication in PLOS ONE. Congratulations! Your manuscript is now with our production department. 

Kind regards, 

on behalf of

Dr. Fiona J Radcliff 

Academic Editor

PLOS ONE